# Active Anti-Inflammatory and Hypolipidemic Derivatives of Lorazepam

**DOI:** 10.3390/molecules24183277

**Published:** 2019-09-09

**Authors:** Panagiotis Theodosis-Nobelos, Georgios Papagiouvannis, Panos N. Kourounakis, Eleni A. Rekka

**Affiliations:** 1Department of Pharmaceutical Chemistry, School of Pharmacy, Aristotelian University of Thessaloniki, 54124 Thessaloniki, Greece (P.T.-N.) (G.P.) (P.N.K.); 2Department of Pharmacy, School of Health Sciences, Frederick University, Nicosia 1036, Cyprus

**Keywords:** non steroidal anti-inflammatory drugs, lorazepam derivatives, antioxidants, inflammation, hyperlipidemia, lipoxygenase inhibition

## Abstract

Novel derivatives of some non steroidal anti-inflammatory drugs, as well as of the antioxidants α-lipoic acid, trolox and (*E*)-3-(3,5-di-tert-butyl-4-hydroxyphenyl)acrylic acid with lorazepam were synthesised by a straightforward method at satisfactory to high yields (40%–93%). All the tested derivatives strongly decreased lipidemic indices in rat plasma after Triton induced hyperlipidaemia. They also reduced acute inflammation and a number of them demonstrated lipoxygenase inhibitory activity. Those compounds acquiring antioxidant moiety were inhibitors of lipid peroxidation and radical scavengers. Therefore, the synthesised compounds may add to the current knowledge about multifunctional agents acting against various disorders implicating inflammation, dyslipidaemia and oxidative stress.

## 1. Introduction

Lorazepam (7-chloro-5-(2-chlorophenyl)-1,3-dihydro-3-hydroxy-2*H*-1,4-benzodiazepin-2-one) is a benzodiazepine derivative, used as anxiolytic, sedative, in status epilepticus and in the treatment of alcohol withdrawal. It is known that benzodiazepines act by potentiating the interaction of GABA with GABA_A_ receptors. Biologic stress has been described as the non-specific adaptive response of the organism to various stimuli, physical, psychological or emotional, such as fear and anxiety [1]. We [1] and others [2] have shown that biologic stress induces oxidative stress, and that GABAergic modulation may offer protection against immobilization-induced stress and oxidative damage [2]. Moreover, it has been reported that the benzodiazepine midazolam protects against neuronal degeneration and apoptosis induced by biological and oxidative stress [3]. Alimentary dyslipidemia disturbed anxiety level and cognitive processes in mice [4] and a diet rich in saturated fat and fructose caused high serum cholesterol and triglyceride concentrations and induced a condition in rats similar to the human metabolic syndrome. This condition was accompanied by anxiety-related behaviour, which correlated significantly with oxidative stress. In addition, the degree of lipid peroxidation correlated well with the metabolic effects of the diet [5]. Finally, a number of benzodiazepine derivatives reduced hyperalgesia in rats in a dose-dependent manner, using the carrageenan-induced method [6].

Taking the above evidence into consideration, in this investigation we report the synthesis and biological evaluation of esters of lorazepam with five classical NSAIDs and the antioxidants α-lipoic acid ((*R*)-5-(1,2-dithiolan-3-yl)pentanoic acid), trolox (6-hydroxy-2,5,7,8-tetramethylchroman-2-carboxylic acid) and (*E*)-3-(3,5-di-*tert*-butyl-4-hydroxyphenyl)acrylic acid (BHA). The aim of this work was to investigate whether such derivatives would combine hypolipidemic and anti-inflammatory activity and to examine their potential lipoxygenase inhibitory, antioxidant and radical scavenging activities. We have demonstrated that a number of lorazepam derivatives possess hypolipidemic action [7]. Furthermore, alpha-lipoic acid reduced inflammation and oxidative stress in patients with metabolic syndrome [8]. We have also reported that trolox and cinnamic acid derivatives demonstrated antioxidant, anti-inflammatory and hypolipidemic activities [9]. Lastly, indole-3-acetic acid, part of the indomethacin structure, was also used. 

## 2. Results and Discussion

### 2.1. Synthesis

Compounds **1**–**6**, **8**, **9** were synthesised by direct esterification of the carboxylic group of the respective acids with lorazepam, using *N*,*N*′-dicyclohexyl-carbodiimide (DCC), at room temperature and high yields. For compound **7**, carbonyldiimidazole (CDI) was used (Figure 1).

### 2.2. Biological Activity

#### 2.2.1. In Vivo Experiments

##### Effect of Compounds on Acute Inflammation in Rats

The effect of the synthesised compounds on acute inflammation, applying the carrageenan rat paw oedema model, as well as the anti-inflammatory activity of the parent NSAIDs, are shown in Table 1.

The carrageenan-induced paw oedema is a commonly and widely used model of acute inflammation. The early phase of carrageenan inflammation is characterised mainly by the release of histamine, serotonin and bradykinin. In the late phase, more than two hours after administration, the additional effects of neutrophil infiltration, prostaglandin production and pro-inflammatory cytokine release develop [10]. In this investigation, oedema was estimated 3.5 h after carrageenan administration.

All compounds demonstrated increased anti-inflammatory activity. NSAID derivatives **1**–**5** were more potent than their individual parent acids. Especially, compound **2** was four times more active than naproxen. The activity of **1** and **5** were about two fold higher than ibuprofen and tolfenamic acid, respectively. Overall, these results indicate a further enhancement of the anti-inflammatory activity by the performed molecular modification. It has been found that diazepam reduces the number of inflammatory cells in the central nervous system [11], treatment with high diazepam doses decreases paw oedema after carrageenan-induced injury [12] and that benzodiazepine derivatives with imide substitution at the C3 of the benzodiazepine ring reduced up to 80% carrageenan rat paw hyperalgesia, in a dose dependent manner, and this activity was at least partly attributed to bradykinin B_1_ receptor antagonism [6].

Compounds **7** and **8** showed considerable anti-inflammatory activity. There is not any reported anti-inflammatory activity for trolox. Additionally, butylated hydroxytoluene (BHT), a well known antioxidant structurally similar to the parent acid ΒHA, lacks such action either in vivo [13], or in vitro [14]. Thus, the antioxidant activity alone does not seem to be entirely responsible for the anti-inflammatory activity. Furthermore, compounds **6** and **9** also inhibited paw oedema, although they do not express any antioxidant activity (part 2.2.2.1.). Indole-3-acetic acid has no reported anti-inflammatory action, while lipoic acid is only a weak anti-inflammatory agent at similar doses [15]. Again, it seems convincing that the lorazepam moiety contributes to increased anti-inflammatory activity.

##### Effect of Compounds on Hyperlipidemia in Rats

The effect of the compounds under investigation on plasma total cholesterol, triglyceride and LDL-cholesterol levels, 24 h post injection, determined in rats after Triton-induced hyperlipidaemia is shown in Table 2. Simvastatin, ibuprofen and naproxen are included for comparison.

Tyloxapol (Triton WR1339) is a nonionic surfactant used to induce hyperlipidemia in experimental animals if administered parenterally. It leads to accumulation of triglycerides, VLDL- and LDL-cholesterol in plasma, reaching maximal effect 24 h after administration. These actions are due to inhibition of lipoprotein lipase and to stimulation of 3-hydroxy-3-methyl-glutaryl-CoA (HMG-CoA) reductase [9,16].

Compounds **1** and **6**–**8** were administered at 0.15mmol/kg and reduced greatly all lipidemic indices, e.g., total cholesterol reduction was at the range of 80%, comparable to that of simvastatin. For most compounds a lower dose, 0.05mmol/kg, was tested and found that still a very significant reduction of lipidemia was achieved, further indicating a dose dependent action. Compounds **1** and **2** were more active than ibuprofen and naproxen at 6–10 times lower dose. These results may be related to their high anti-inflammatory activity and partly may be related to a potential antioxidant effect.

#### 2.2.2. In Vitro Experiments

##### Antioxidant Activity

Considering that free radicals are implicated in inflammatory processes, the synthesised compounds were tested for antioxidant activity, expressed as inhibition of rat microsomal membrane lipid peroxidation induced by ferrous ascorbate, as well as interaction with 1,1-diphenyl-2-picrylhydrazyl radical (DPPH). The percent interaction with DPPH and the IC_50_ values of the active final compounds and trolox from rat hepatic microsomal membrane lipid peroxidation, after 45 min of incubation are shown in Table 3.

The time course of lipid peroxidation inhibition, as affected by various concentrations of **7** is shown in Figure 2.

Most compounds, except for **7** and **8**, were practically inactive in these experiments. In the lipid peroxidation test, compound **8** showed moderate activity. This may be due to the high lipophilicity of compound **8** (logP = 7.4) which reduced its solubility in the aqueous reaction environment. However, it interacted strongly with DPPH, in a way comparable to that of trolox. Compound **7** was almost tenfold more potent inhibitor of lipid peroxidation than trolox, a reference antioxidant, and an effective radical scavenger. It is possible that the hypolipidemic and anti-inflammatory effects of these compounds are related, at least partly, to their antioxidant activity. The lipoic acid derivative **6** was found inactive, and this is in accordance with the observation that lipoic acid can scavenge only very reactive radicals, while the reduced form, 6,8-dimercaptooctanoic acid, is a strong antioxidant due to hydrogen transfer [17,18]. Thus, it could be suggested that the reduced form of **6** may contribute to the in vivo anti-inflammatory and hypolipidemic activity of this compound.

##### Inhibition of Lipoxygenase

Lipoxygenases, involved in inflammation, are a family of enzymes that catalyse the dioxygenation of polyunsaturated fatty acids which contain the *cis*-1,4-pentadiene structure. Although there are several enzymes, they all catalyse the stereo- and regio-specific peroxidation of arachidonic or linoleic acid in the presence of molecular oxygen. Soybean lipoxygenase-1 can use arachidonic acid as substrate, with about 15% of activity for linoleic acid. It has been found that arachidonic acid binding sites in plant lipoxygenases share almost the same similarity with animal 5-lipoxygenase [19]. 5-Lipoxygenase activity contributes to atherosclerosis via oxidation of low-density lipoprotein. Furthermore, studies using 5-lipoxygenase-deficient mice show that 5-lipoxygenase activity may contribute to stress and depression behaviour [20].

The effect of the synthesised compounds on soybean lipoxygenase-1, using linoleic acid as substrate, is demonstrated in Table 4. In this table, the inhibition offered by nordihydroguaiaretic acid (NDGA), a potent lipoxygenase inhibitor, is included. BHA and trolox could not inhibit lipoxygenase even at concentrations much higher than 300 µΜ. In addition, no inhibition was observed when linoleic acid was used at 1 mM, a concentration higher than the saturating substrate concentration, under the same experimental conditions. The decline of inhibition by increasing the concentration of the substrate indicates a competitive inhibition of lipoxygenase.

The time course of lipoxygenase inhibition by the most active of the synthesised compounds, **3** and **8**, is shown in Figure 3.

From the presented results it could be observed that, when lorazepam was esterified with rigid acids, e.g., naproxen (**2**), trolox (**7**), indole-3-acetic acid (**9**), inhibition was insignificant or absent, whereas, less rigid substitution seems to contribute to stronger inhibition, e.g., ketoprofen (**3**), lipoic acid (**6**), BHA (**8**).

## 3. Materials and Methods

### 3.1. General 

All commercially available reagents were purchased from Merck (Kenilworth, NJ, USA) and used without further purification. κ-Carrageenan and lipoxygenase type I from soybean were purchased from Sigma (St. Louis, MO, USA). The IR spectra were recorded on a Perkin Elmer Spectrum BX FT-IR spectrometer (Waltham, MA, USA). The ^1^H-NMR spectra were recorded using an AGILENT DD2-500 MHz spectrometer (Santa Clara, CA, USA). All chemical shifts are reported in δ (ppm) and signals are given as follows: s, singlet; d, doublet; t, triplet; m, multiplet. Melting points (m.p.) were determined with a MEL-TEMPII Laboratory Devices, Sigma-Aldrich (Milwaukee WI, USA) apparatus and are uncorrected. The microanalyses were performed on a Perkin-Elmer 2400 CHN elemental analyser. Wistar rats (160–220 g, 3–4 months old) were kept in the Centre of the School of Veterinary Medicine (EL54 BIO42), Aristotelian University of Thessaloniki, which is registered by the official state veterinary authorities (presidential degree 56/2013, in harmonization with the European Directive 2010/63/EEC). The experimental protocols were approved by the Animal Ethics Committee of the Prefecture of Central Macedonia (no. 270079/2500).

### 3.2. Synthesis

#### General Method for the Synthesis of Compounds **1**–**9**

A) In a solution of the corresponding acid (1mmol) in dichloromethane (CH_2_Cl_2_) lorazepam is suspended (1.05 mmol) and *N*,*N*′-dicyclohexylcarbodiimide (DCC, 1.3mmol) was added. The reaction mixture was stirred for 4–12h. After filtration, the final compounds were isolated with flash chromatography using petroleum ether and ethyl acetate as eluents.

B) For compound **7**: Trolox (6-hydroxy-2,5,7,8-tetramethylchroman-2-carboxylic acid, 1mmol) was dissolved in tetrahydrofuran (THF) and carbonyldiimidazole (CDI, 1.2 mmol) was added. After stirring for 45 min, lorazepam (1.05 mmol) was added and the reaction was left for 12 h at room temperature. The solvent was distilled off and the residue was dissolved in ethyl acetate and washed with water. The solution was dried over Na_2_SO_4_ and the final compound was isolated with flash chromatography using petroleum ether and ethyl acetate as eluents. 

7-Chloro-5-(2-chlorophenyl)-2-oxo-2,3-dihydro-1*H*-benzo[e][1,4]diazepin-3-yl 2-(4-isobutylphenyl) propanoate (**1**). Flash chromatography (petroleum ether/ethyl acetate, 4/1). White solid, yield 74%, m.p. 105–122 °C. IR (nujol): 3297 (N-H), 1738, 1714 (C=O ester diastereomers), 1621 (C=O amide), 1527 (C-C aromatic) cm^−1^. ^1^H-NMR (CDCl_3_) *δ*: 0.92 (d, 6H, *J* = 6.4 Hz, CH_3_-CH-CH_3_), 1.65 (dd, 3H, *J* = 14.1, 7.1 Hz, -CO-CH-CH_3_), 1.95–1.78 (m, 1H, CH_3_-CH-CH_3_), 2.46 (d, 2H, *J* = 7.1 Hz, -CH_2_CH-(CH_3_)_2_), 4.05 (q, 1H, *J* = 7.1 Hz, -CO-CH-CH_3_), 6.02, 6.00 (s, 1H, -O-CH-C=O), 7.17–7.00 (m, 4H, aromatic ibuprofen), 7.65–7.30 (m, 7H, aromatic lorazepam), 9.21 (s, 1H, -NH). Anal. Calcd for C_28_H_26_Cl_2_N_2_O_3_: C, 66.02; H, 5.14; Ν, 5.50. Found: C, 65.82; H, 5.30; N, 5.33.

7-Chloro-5-(2-chlorophenyl)-2-oxo-2,3-dihydro-1*H*-benzo[e][1,4]diazepin-3-yl 2-(6-methoxynaphthalen-2-yl)propanoate (**2**). Flash chromatography (petroleum ether/ethyl acetate, 3/1). White solid, yield 93%, m.p. 136–139 °C. IR (nujol): 3220 (N-H), 1702 (C=O ester), 1627 (C=O amide), 1605, 1569 (C-C aromatic) cm^−1^. ^1^H-NMR (CDCl_3_) *δ*: 1.75 (d, 3H, *J* = 7.7 Hz, CH_3_-CH-C=O), 3.94 (s, 3H, -O-CH_3_), 4.28–4.10 (m, 1H, CH_3_-CH-C=O), 6.01 (s, 1H, -O-CH-C=O), 6.99 (d, 1H, *J* = 8.5 Hz, naphthyl C2), 7.04 (s, 1H, naphthyl C10), 7.18–7.08 (m, 2H, naphthyl C7, C8), 7.45–7.25 (m, 3H, chlorophenyl C3, C4, C5 and naphthyl C5), 7.52 (d, 1H, *J* = 8.5 Hz, naphthyl C3), 7.76–7.55 (m, 3H, chlorophenyl C6 and chlorophenylamino C3, C5), 7.83 (d, 1H, *J* = 8.1 Hz, chlorophenylamino C6), 9.42 (s, 1H, O=C-NH-). Anal. Calcd for C_29_H_22_Cl_2_N_2_O_4_: C, 65.30; H, 4.16; Ν, 5.25. Found: C, 65.42; H, 4.26; N, 4.89.

7-Chloro-5-(2-chlorophenyl)-2-oxo-2,3-dihydro-1*H*-benzo[e][1,4]diazepin-3-yl 2-(3-benzoylphenyl)propanoate (**3**). Flash chromatography (petroleum ether/ethyl acetate, 3/1).White solid, yield 85%, m.p.195–201 °C. IR (nujol): 3203 (N-H), 1739 (C=O ketone), 1712, 1696 (C=O ester diastereomers), 1650 (C=O amide), 1596 (C-C aromatic) cm^−1^. ^1^H-NMR (CDCl_3_ + DMSO-*d*_6_) *δ*: 1.73–1.67 (m, 3H, -CO-CH-CH_3_), 4.20–4.08 (m, 1H, -CO-CH-CH_3_), 6.04, 6.02 (s, 1H, -O-CH-C=O), 7.15–7.00 (m, 2H, isopropylphenyl C5, chlorophenyl C5), 7.93–7.33 (m, 14H, aromatic), 8.70, 8.73 (s, 1H, NH-). Anal. Calcd for C_31_H_22_Cl_2_N_2_O_4_: C, 66.80; H, 3.98; Ν, 5.03. Found: C, 66.54; H, 3.91; N, 4.67.

7-Chloro-5-(2-chlorophenyl)-2-oxo-2,3-dihydro-1*H*-benzo[e][1,4]diazepin-3-yl 2-(1-(4-chlorobenzoyl)-5-methoxy-2-methyl-1H-indol-3-yl)acetate (**4**). Flash chromatography (petroleum ether/ethyl acetate, 2/1). White solid, yield 81%, m.p. 189–191 °C. IR (nujol): 3302 (N-H), 1748 (C=O ester), 1699 (C=O amide indole), 1620 (C=O amide lorazepam), 1574 (C-C aromatic) cm^−1^. ^1^H-NMR (CDCl_3_ + DMSO-*d*_6_) *δ*: 2.25 (s, 3H, CH_3_-C-N-), 3.71 (s, 3H, -O-CH_3_), 3.83 (s, 2H, -CH_2_-C=O), 5.84 (s, 1H, -O-CH-C=O), 6.55 (d, 1H, *J* = 9.0 Hz, indole C6), 6.85 (d, 1H, *J* = 9.0 Hz, indole C7), 6.89 (s, 1H, indole C4), 6.98 (s, 1H, chlorophenylamino C3), 7.13 (d, 1H, *J* = 8.7 Hz, chlorophenyl C3), 7.31–7.21 (m, 4H, chlorophenylamino C5 and chlorophenyl C4, C5, C6), 7.33 (d, 2H, *J* = 8.4 Hz, aromatic chloro-benzoyl C3, C5), 7.41 (d, 1H, *J* = 6.0 Hz, chlorophenylamino C6), 7.54 (d, 2H, *J* = 8.4 Hz, chlorobenzoyl C2, C6), 10.61 (s, 1H, -NH-). Anal. Calcd for C_34_H_24_Cl_3_N_3_O_5_: C, 61.79; H, 4.56; Ν, 6.36. Found: C, 61.65; H, 4.22; N, 5.92.

7-Chloro-5-(2-chlorophenyl)-2-oxo-2,3-dihydro-1*H*-benzo[e][1,4]diazepin-3-yl 2-((3-chloro-2-methylphenyl)amino)benzoate (**5**). Flash chromatography (petroleum ether/ethyl acetate, 2/1). White solid, yield 41%, m.p. 206–207 °C. IR (nujol): 3327 (N-H), 1721 (C=O ester), 1635 (C=O amide), 1583 (C-C aromatic) cm^−1^. ^1^H-NMR (CDCl_3_ + DMSO-*d*_6_) *δ*: 2.25 (s, 3H, -C**H_3_**), 6.14 (s, 1H, -O-C**H**-C=O), 6.80–6.67 (m, 2H, aromatic amino methyl phenyl C6 and aromatic benzoic acid C5), 7.45–6.96 (m, 10H, aromatic chloro-methyl-phenylamino C4, C5, benzoic acid C3, C4, C6, chlorophenylamino C3, C5 and chlorophenyl C3, C4, C5), 7.55 (d, 1H, *J* = 7.6 Hz, chlorophenyl C6), 8.25 (d, 1H, *J* = 8.1 Hz, chlorophenylamino C6), 9.09 (s, 1H, -N**H**-), 10.93 (s, 1H, O=C-NH-). Anal. Calcd for C_29_H_20_Cl_3_N_3_O_3_: C, 61.66; H, 3.57; Ν, 7.44. Found: C, 62.03; H, 3.53; N, 7.81.

7-Chloro-5-(2-chlorophenyl)-2-oxo-2,3-dihydro-1*H*-benzo[e][1,4]diazepin-3-yl 5-(1,2-dithiolan-3-yl)pentanoate (**6**). Flash chromatography (petroleum ether/ethyl acetate, 1/1). White solid, yield 71%, m.p. 184–189 °C. IR (nujol): 3237 (N-H), 1713, 1682 (C=O ester), 1620 (C=O amide), 1592 (C-C aromatic) cm^−1^. ^1^H-NMR (CDCl_3_ + DMSO-*d*_6_) *δ*: 1.80–1.67 (m, 2H, C3 H), 1.62–1.47 (m, 4H, C4, C5), 1.93–1.84 (m, 1H, C7 axial), 2.33–2.21 (m, 1H, C7 equatorial), 2.43–2.35 (m, 2H, C2), 3.03–2.86 (m, 2H, C6, C8 axial), 3.42–3.32 (m, 1H, C8 equatorial), 5.75 (s, 1H, -O-CH-C=O), 6.83 (d, 1H, *J* = 2.3 Hz, chlorophenyl C3), 7.09 (d, 1H, *J* = 8.7 Hz, chlorophenyl C6), 7.28–7.17 (m, 4H, chlorophenylamino C3, C5 and chlorophenyl C4, C5), 7.39–7.34 (m, 1H, chlorophenylamino C6), 10.64 (s, 1H, O=C-NH-). Anal. Calcd for C_23_H_22_Cl_2_N_2_O_3_S_2_: C, 54.22; H, 4.35; Ν, 5.50. Found: C, 54.41; H, 4.68; N, 5.33.

7-Chloro-5-(2-chlorophenyl)-2-oxo-2,3-dihydro-1*H*-benzo[e][1,4]diazepin-3-yl 6-hydroxy-2,5,7,8-tetramethylchroman-2-carboxylate (**7**). Flash chromatography (petroleum ether/ethyl acetate, 3/2). White solid, yield 85%, m.p. 138–144 °C. IR (nujol): 3400 (O-H), 3250 (N-H), 1680 (C=O ester), 1631 (C=O amide) cm^−1^. ^1^H-NMR (CDCl_3_) *δ*: 1.78 (s, 3H, 2-CH_3_), 2.00 (s, 3H, 5-CH_3_), 2.10 (s, 3H, 7-CH_3_), 2,15 (s, 3H, 8–CH_3_), 2.20–2.65 (m, 4H, chromane), 5.10 (s, 1H, -OH), 6.10 (s, 1H, -O-CH-C=O), 7.20 - 7.82 (m, 7H, aromatic), 9.40 (s, 1H, -NH-). Anal. Calcd for C_29_H_26_Cl_2_N_2_O_5_: C, 62.94; H, 4.74; Ν, 5.06. Found: C, 62.69; H, 4.96; N, 5.13.

(*E*)-7-Chloro-5-(2-chlorophenyl)-2-oxo-2,3-dihydro-1*H*-benzo[e][1,4]diazepin-3-yl 3-(3,5-di-*tert*-butyl-4-hydroxyphenyl)acrylate (**8**). Flash chromatography (petroleum ether/ethyl acetate, 6/1 and then 3/1). White solid, yield 40%, m.p. 274 °C. IR (nujol): 3611 (O-H), 3277 (N-H), 1710 (C=O ester), 1627 (C=O amide), 1593 (C-C aromatic) cm^−1^. ^1^H-NMR (CDCl_3_ + DMSO-*d*_6_) *δ*: 1.49 (s, 18H, -CH_3_), 5.56 (s, 1H, phenol OH), 6.00 (s, 1H, -O-CH-C=O), 6.61 (d, 1H, *J* = 15.9 Hz, CH=CH-C=O), 7.89 (d, 1H, *J* = 15.9 Hz, CH=CH-C=O), 7.13 (d, 1H, *J* = 2.2 Hz, chlorophenyl C6), 7.21 (d, 1H, *J* = 8.7 Hz, chlorophenyl C3), 7.51–7.39 (m, 6H, chlorophenyl C4, C5, chlorophenylamino C3, C5 and di-*tert*-butylphenyl C2, C6), 7.64 (d, 1H, *J* = 7.4 Hz, chlorophenylamino C6), 7.89 (d, 1H, *J* = 15.9 Hz, CH=CH-C=O), 8.97 (s, 1H, -NH-). Anal. Calcd for C_32_H_32_Cl_2_N_2_O_4_ x0.7CH_2_Cl_2_: C, 61.47; H, 5.27; Ν, 4.38. Found: C, 61.31; H, 5.58; N, 4.06.

7-Chloro-5-(2-chlorophenyl)-2-oxo-2,3-dihydro-1*H*-benzo[e][1,4]diazepin-3-yl 2-(1*H*-indol-3-yl)acetate (**9**). Flash chromatography (petroleum ether/ethyl acetate, 2/1). White solid, yield 40%, m.p. 257–258 °C. IR (nujol): 3360 (N-H), 1760 (C=O ester), 1713 (C=O amide), 1621 (C-C aromatic) cm^−1^. ^1^H-NMR (CDCl_3_ + DMSO-d_6_) *δ*: 3.96 (s, 1H, -CH_2_-C=O), 5.89 (s, 1H, -O-CH-C=O), 6.94 (d, 1H, *J* = 2.2 Hz, indole C7), 6.99 (d, 1H, *J* = 7.5 Hz, indole C6), 7.06 (t, 1H, *J* = 7.5 Hz, indole C6), 7.25–7.17 (m, 2H, aromatic indole C2, C4), 7.59–7.28 (m, 7H, aromatic lorazepam), 10.16 (s, 1H, indole -NH-), 10.89 (s, 1H, O=C-NH-). Anal. Calcd for C_25_H_17_Cl_2_N_3_O_3_ x2.4H_2_O: C, 57.57; H, 4.21; Ν, 8.06. Found: C, 57.53; H, 3.88; N, 8.12.

### 3.3. Effect on Carrageenan-Induced Rat Paw Oedema

An aqueous solution of carrageenan was prepared (1% *w/v*) and 0.1 mL of this was injected *i.d.* into the right hind paw of female rats; the left paw served as control. The tested compounds (suspended in water with a few drops of Tween 80) were given *intra peritoneally* (*i.p.*) (0.15 mmol/kg) 5 min prior to the carrageenan injection. After 3.5 h the hind paws were weighed separately. The produced oedema was estimated as paw weight increase [18].

### 3.4. Effect on Plasma Total Cholesterol, Triglyceride and LDL-cholesterol Levels

A solution of Triton WR 1339 (tyloxapol) in saline was administered *i.p.* (200 mg/kg) to male rats and 1 h later the examined compound (0.15 and/or 0.05 mmol/kg, suspended in water with a few drops of Tween 80) was given *i.p*. After 24 h, blood was taken from the aorta and used for the determination of plasma total cholesterol (TC), triglyceride (TG) and low density lipoprotein cholesterol (LDL-C) concentrations, using commercial kits, against standard solutions [21].

### 3.5. Effect on Lipid Peroxidation

The incubation mixture contained heat-inactivated rat hepatic microsomal fraction, ascorbic acid (0.2 mM) in Tris–HCl/KCl buffer (pH 7.4) and the test compounds dissolved in dimethylsulphoxide. The reaction was initiated by FeSO_4_ (10 µM) and the mixture was incubated at 37 °C. Aliquots were taken at various time intervals for 45 min. Lipid peroxidation was assessed spectrophotometrically (535 against 600 nm) by the determination of 2-thiobarbituric acid (TBA) reactive material [22].

### 3.6. Interaction with the Stable Radical 1,1-diphenyl-2-picrylhydrazyl (DPPH)

Compounds (in absolute ethanol, final concentration 50–200 µM) were added to an equal volume of an ethanolic solution of DPPH (final concentration 200 µM) at room temperature (22 ± 2 °C). Absorbance (517 nm) was recorded after 30 min [22].

### 3.7. Effect on Lipoxygenase Activity

The reaction mixture contained the test compounds dissolved in ethanol and soybean lipoxygenase (250 U/mL) in Tris buffer (pH 9). The reaction was initiated by the addition of sodium linoleate (0.1 mM) and monitored for 7 min at 28 °C, by recording the absorbance of a conjugated diene structure at 234 nm. For the estimation of the type of inhibition, the above experiments were repeated, using sodium linoleate at 1 mM, which is higher than the saturating substrate concentration [18].

## 4. Conclusions

In conclusion, the synthesised compounds presented important hypolipidemic activity and significant in vivo anti-inflammatory properties, being superior to the parent acids. It is reported that anxiolytic therapy improves stress-induced inflammation [23]. Furthermore, inflammation, endothelial dysfunction and platelet aggregation may connect anxiety disorders with cardiovascular diseases [24]. Finally, we have shown that molecules containing lorazepam and antioxidants, linked by a GABA residue, reduced stress, caused by immobilization and fasting, as well as the subsequent oxidative damage [7]. Complex stress-related disorders could be treated effectively with agents designed to act at different stages of their pathogenesis. The present research may provide useful candidate molecules towards this target.

## Figures and Tables

**Figure 1 molecules-24-03277-f001:**
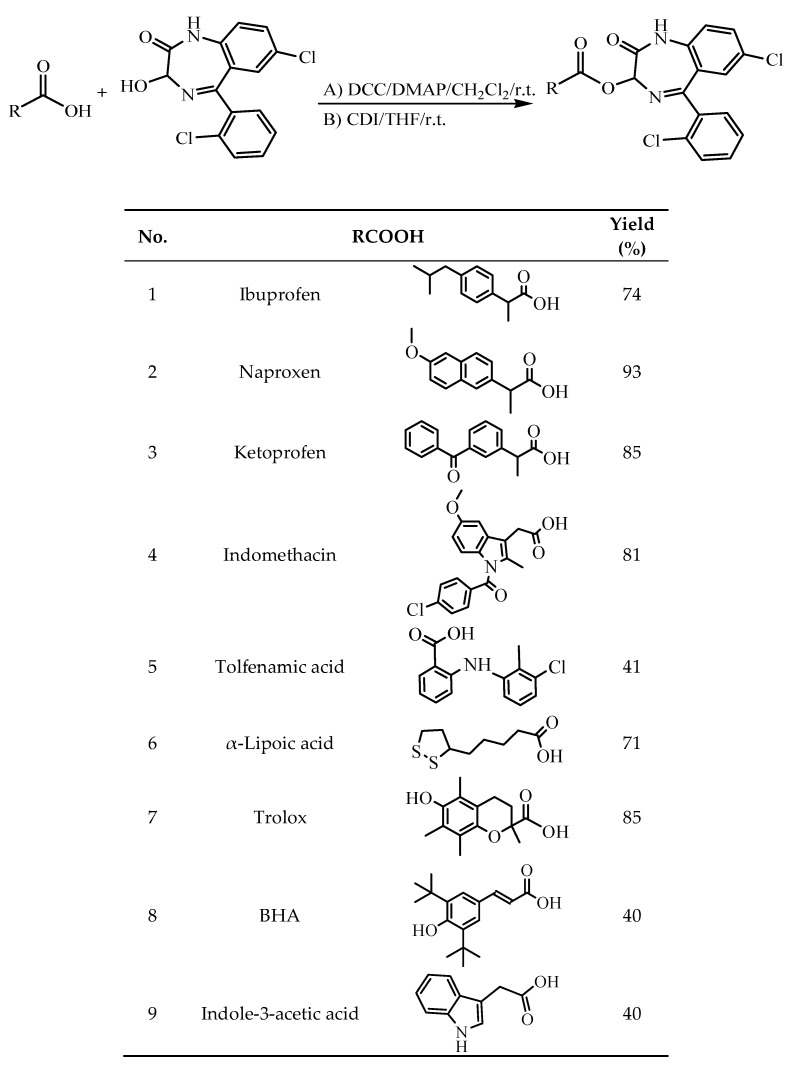
Synthesis and structures of the examined compounds. r.t.

**Figure 2 molecules-24-03277-f002:**
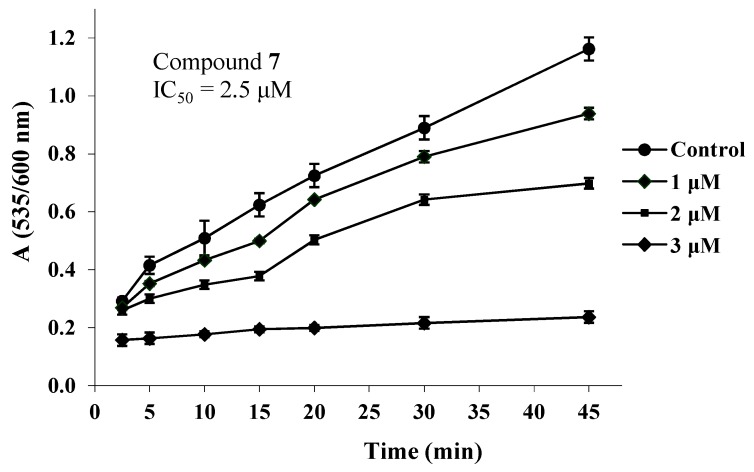
Inhibition of lipid peroxidation, as affected by various concentrations of **7**, in relation to time.

**Figure 3 molecules-24-03277-f003:**
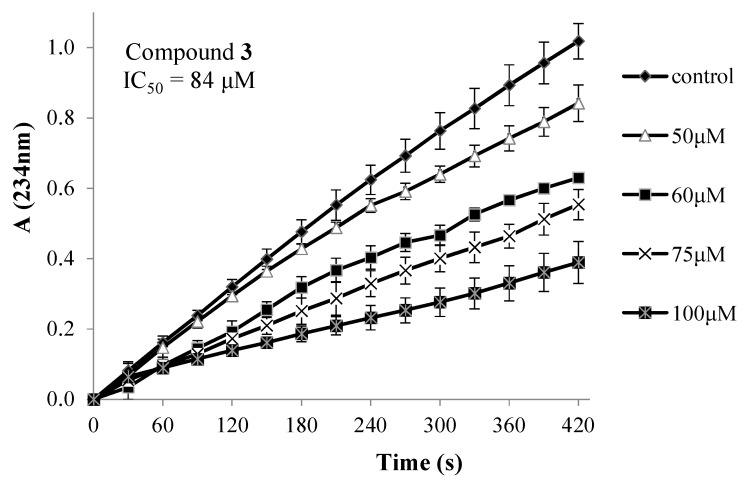
Time course of lipoxygenase inhibition by various concentrations of compounds **3** and **8**.

**Table 1 molecules-24-03277-t001:** Effect of compounds **1**–**9**, ibuprofen, naproxen, ketoprofen, indomethacin and tolfenamic acid on carrageenan-induced rat paw oedema ^a.^

Compound	% Oedema Reduction
**1**	68 **
Ibuprofen	36 *
**2**	46 **
Naproxen	11 *
**3**	65 **
Ketoprofen	47 *
**4**	67 **
Indomethacin	42 **
**5**	55 **
Tolfenamic acid	24 **
**6**	39 *
**7**	56 *
**8**	37 *
**9**	43 **

^a^ The effect on oedema is expressed as percent inhibition of oedema in comparison to controls, 3.5 h post administration. All compounds were administered *intra peritonealy* (*i.p.*) at 0.15 mmol/kg body weight. Significant difference from control: * *p* < 0.01, ** *p* < 0.001 (Student’s *t* test).

**Table 2 molecules-24-03277-t002:** Effect of the tested compounds, simvastatin, ibuprofen and naproxen on Triton WR1339 (tyloxapol) induced hyperlipidemia.

Compound	Dose *i.p.* (µmol/kg)	% Reduction		
		TC ^a^	TG ^b^	LDL-C ^c^
**1**	150	82 ***	65 ***	56 *
**1**	50	44 ***	57 **	43 **
**2**	50	60 ***	71 ***	42 ***
**4**	50	59 ***	59 ***	60 ***
**5**	50	56 ***	57 ***	48 ***
**6**	150	82 ***	41 **	60 *
**6**	50	48 ***	39 ***	47 ***
**7**	150	72 ***	64 *	69 *
**8**	150	81 ***	66 ***	63 ***
**8**	50	59 ***	52 ***	44 **
Simvastatin	150	73 ***	-	70 ***
Ibuprofen	300	41 ***	38 ***	42 ***
Naproxen	500	53 ***	44 ***	26 ***

^a^ TC: Total cholesterol; ^b^ TG: Triglycerides; ^c^ LDL-C: LDL cholesterol. Tyloxapol: 200 mg/kg, *i.p*. Significant difference from hyperlipidemic control: * *p* < 0.01, ** *p* < 0.005, *** *p* < 0.001 (Student’s *t* test).

**Table 3 molecules-24-03277-t003:** Interaction of compounds **7**, **8** and trolox, at various concentrations, with DPPH (200 µΜ) ^a^ and their effect on lipid peroxidation ^b.^

Compound	Percent Interaction with DPPH			Inhibition of Lipid Peroxidation IC_50_ (µΜ)
	200 µΜ	100 µΜ	50 µΜ	
**7**	91	82	46	2.5
**8**	90	49	30	> 100
Trolox	92	90	38	25

^a^ After 30 min of incubation. ^b^ After 45 min of incubation. Trolox: 6-hydroxy-2,5,7,8-tetramethylchroman-2-carboxylic acid. All determinations were performed at least in triplicate and standard deviation is always within ± 10% of the mean value.

**Table 4 molecules-24-03277-t004:** Effect of compounds **1**–**9,** BHT, ibuprofen, ketoprofen, tolfenamic acid and NDGA on lipoxygenase ^a.^

Compound	IC_50_ (µΜ) or % Inhibition/µΜ
**1**	14%/50 µΜ
**2**	-
**3**	84
**4**	86
**5**	22%/100 µΜ
**6**	60
**7**	-
**8**	44
**9**	255
BHT	192
Ibuprofen	200
Ketoprofen	220
Tolfenamic acid	170
NDGA	1.3

^a^ After 7 min of incubation; BHT: 2,6-di-*tert*-butyl-4-methylphenol (butylated hydroxytoluene); NDGA: nordihydroguaiaretic acid; -: inactive.

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
