# Peer review of "Active Anti-Inflammatory and Hypolipidemic Derivatives of Lorazepam"

_molecules, 2019, doi:10.3390/molecules24183277_

Round 1

Reviewer 1 Report

In this study synthesis and activity studies of esters of lorazepam with five classical NSAIDs and the antioxidants α-lipoic acid, pentanoic acid, trolox, carboxylic acid and BHA. Althought the number of the compounds are not satisfactory, the aim and the idea of the work is good. At the end of the experiments there are  some significant  combine hypolipidemic and antiinflammatory activity as well as lipoxygenase inhibitory, antioxidant and radical scavenging activities were observed.

There are some additional information for the introduction and discussion parts are definitely needed. For the conclusion authors needs to compare the results with similar studies and compounds. Also some structure activity relationship is needed to explane the activity results.

The Figure 1 is not understable. It should be like a table showing the R groups. Some moderate language corrections are needed.

Reviewer 2 Report

The manuscript: Active anti-inflammatory and hypolipidemic derivatives of lorazepam by E. Rekka et all describes ester derivatives of lorazepam with anti-inflammatory and lipid lowering properties. The chemistry is straight forward and the products are fully characterized.

There are a few questions to the authors: What are the effects of these products on HDL compared with the statin?

The authors describe the effects on lipoxidase, but the use of soybean lipoxidase converts AA to 15-Hpete, but does not produce leukotrienes. Only the 5-Lipoxidase available from Sigma or Cayman produce the precursor of leukotrienes 5-Hpete. This part of the text has to be revised.

Last question; What are the effects of giving the compounds after the carrageenan-induced paw oedema was induced compared with control.
